

# On the use of high resolution satellite imagery to estimate irrigation volumes and its impact in land surface modeling

Jordi Etchanchu[1], Vincent Rivalland[1], Stéphanie Faroux[2], Aurore Brut[1], Gilles Boulet[1]

[1] *CESBIO, Université de Toulouse, CNES, CNRS, IRD, UPS, INRA, Toulouse, France*
[2] *CNRM-GAME, Météo France, CNRS, Toulouse, France*

*Correspondence to*: Jordi Etchanchu (jordi.etchanchu@cesbio.cnes.fr)

**Abstract.** Irrigation is a major issue for water resources management agencies as it is the main component of human fresh water consumption. However, irrigation can be monitored at plot scale but not at larger scales, i.e. from river basin to global scale. Hence, simulating the irrigation process in models is of great interest, not only to forecast the water availability, but
also to provide realistic lower boundary conditions for atmosphere and climate models. This process is relatively well represented in agronomical or agro-hydrological models, designed for crop and water management at the plot scale. But this kind of model is not adapted for water management at the basin scale or even larger scale, due to their complexity. Land Surface Models (LSMs) are used for this purpose. However, irrigation is not well represented in LSMs. These models use basic decision rules to estimate irrigation volumes. Most of the time, it only consists in triggering an irrigation event when
the soil moisture in the root zone drops below a fixed threshold. This threshold is unique at global scale, being independent of the crop type or the common irrigation practices in the simulated area. Then an irrigation amount is applied based on the volume needed to replenish the soil reservoir to a fixed level. There is no consideration about actual agricultural practices. These simple irrigation schemes do not have the flexibility needed to adapt to the wide variety of crops and irrigation practices encountered at large scales. The present study aims at developing and evaluating an irrigation scheme very similar
to the one used in agronomical or agro-hydrological models for the SURFEX-ISBA LSM developed by Meteo-France. Particularly, it allows adapting the triggering threshold spatially and temporally and relating it to the actual phenology of the crop and to the irrigation practices. But increasing the flexibility of a model also means that it needs more input information to constrain it. High-resolution remote sensing products, like those derived from Sentinel-2, can provide part of this information spatially. This study thus presents a method to determine irrigation parameters, and particularly the triggering
soil moisture threshold, from high-resolution remotely sensed leaf area index. This method is compared to three other experiments: a reference simulation with the current irrigation scheme of SURFEX-ISBA, a second experiment designed to show the contribution of remotely sensed irrigation period determination in the current scheme and a third which uses a single threshold over the season. The comparison is done on several maize plots in southwestern France. The results show that the method using remote sensing to modulate the triggering soil moisture threshold shows the best performances in
estimating annual irrigation volumes. Indeed, it shows a bias around 10 mm per year and a RMSE around 30 mm whereas the standard scheme shows a bias around 50 mm per year and a RMSE around 60 mm.. The sensitivity to the estimation of





the soil maximal available water content is then performed. It shows that all the experiments are very sensitive when the maximal available water content in the soil is low. Finally, the impact on evapotranspiration is evaluated. It shows small differences between experiments and with the measured evapotranspiration. This study thus shows the potential of using high resolution remote sensing products to improve the irrigation simulation in LSMs. Indeed, it allows increasing the

realism of the irrigation scheme while keeping it generic enough to simulate at regional to global scale.

## 1 Introduction

About 70% of the global freshwater consumption is used for agriculture, mainly for irrigation (WWAP, 2014). But irrigation is most often needed in periods with a relatively high water stress and low water availability. It is thus critical for water management agencies to assess the irrigation water use and its effects on the water resources availability. Nowadays Land

Surface Models (LSM) are often used to forecast the water resources availability (Habets et al., 2008; Tesemma et al., 2015). Hence, many developments in LSMs aim at integrating anthropogenic effects like irrigation in the simulation of surface water and energy fluxes. By coupling these models with atmospheric or hydrological models, several studies evaluated the effects of irrigation on the water cycle. Haddeland et al. (2006) showed that irrigation can significantly impact river discharge. This effect is mainly due to the water uptake in the rivers themselves, but also reservoirs or aquifers (Leng et al,

2013, 2017). Irrigation also impacts the hydrometeorological fluxes and therefore the surface-atmosphere coupling. Particularly, it increases the latent heat flux and decreases the sensible heat flux compared to rain-fed crops (De Rosnay et al., 2003; Lobell et al., 2009, Qian, 2013). Consequently, there is a cooling effect of the irrigation as well as an increase of the humidity near the surface (Zaitchik et al., 2005; Ozdogan et al., 2010; Lawtson et al., 2015). Those studies show that modeling realistic irrigation practices is a major issue not only to monitor the water resources and assess its future

availability, but also to provide realistic lower boundary conditions for the atmosphere.

Some LSMs actually use fairly simple models, with few parameters, to estimate the irrigation volumes. The most common way used to trigger irrigation events in these models is to set a soil moisture threshold. When the soil moisture drops below this threshold, an irrigation event is triggered (Evans and Zaitchik, 2008; Lobell et al. 2009, Kueppers and Snyders, 2012, Williams et al., 2016). In these studies, the threshold was fixed to a constant value and do not take into account the crop type,

the actual phenology of the plant or the irrigation behaviors of the farmers. Some studies include phenological considerations by using a triggering threshold on evapotranspiration (De Rosnay et al., 2003) or linking the soil moisture threshold to the stress level of the different crops (Haddeland et al., 2006; Pokhrel et al., 2011; Leng et al., 2013). But none of these schemes has the flexibility to account for farmers management. Indeed, the farmers modulate the irrigation triggering level depending on the crop types and growing stages but also knowing the constraints on their irrigation material. In these studies, the

irrigation amount for each event is then determined by evaluating the water needed to replenish the soil to its field capacity (Haddeland et al., 2006, Evans and Zaitchik, 2008, Willimas et al., 2016) or to a level where the plant is assumed to avoid water stress (Lobell et al., 2009; Pokhrel et al., 2011, Leng et al., 2013).). Such determination of the irrigation amount is not



realistic. In fact, for practical reasons, irrigation systems generally provide a constant amount of water depending on their capacity, for all irrigation events. This value should not be determined based on the water deficit in the soil but prescribed in accordance with the local irrigation practices. Some of these studies also do not account for an explicit irrigation season (De Rosnay et al., 2003, Lobell et al., 2009, Leng et al., 2013). So they rely on the assumption that the irrigation will trigger only

in the cropping period when the soil moisture is the most likely to drop under the prescribed threshold. The others base their irrigation season on climatologic values (Evans et Zaitchik, 2008, Ozdogan et al., 2010, Pokhrel et al., 2011). But the irrigation season must be adapted to the earliness of each crop variety.

Agro-hydrological models, like SAFYE (Battude et al., 2017) or SAMIR (Saadi et al., 2015), are meant for water management and farmer advice at the farm scale. Their irrigation schemes explicitly account for irrigation management

rules. Particularly, it allows the user to prescribe the irrigation period and the amount for each event. They also include rules inherent to the irrigation material or the crop management, like the minimal return period between two irrigation events or the drying period before harvest for cereals. Hence, these irrigation schemes solve most of the problems pointed out on LSMs formalisms. However, they also used fixed irrigation threshold. This means that the irrigation practice cannot be modulated according to the crop growth stage. Agronomic models like STICS (Brisson-Cohen et al., 2009) or MAELIA

(Therond et al., 2014) offer this possibility. But using such models at large scale is not possible due to their complexity and the number of input data they need. Hence, the present study aims at developing and evaluating an irrigation scheme for the SURFEX-ISBA (Noilhan et Planton, 1989, Masson et al., 2013) land surface model. It is meant to share the same complexity as SAFYE and SAMIR with the possibility to distribute some of the parameters in space and time. This more flexible irrigation scheme could thus be adapted to the wide variety of crops and irrigation practices encountered when simulating at

regional scales without reaching the complexity of the agronomical models.

However, distributing the model parameters in space and time also means that relevant spatio-temporal information is needed to drive this distribution. The studies mentioned above generally use statistics on irrigated areas from the Food and Agriculture Organization (Siebert et al., 2013) to simulate regional to global scales. These statistics have coarse resolution and cannot represent accurately the spatial distribution of the irrigated plots, especially in areas where the plots are small like

in Western Europe. Hence, there are huge uncertainties on the location of the irrigation events. The temporality of the events is also impacted since the simulated soil water content at such resolution may not be representative of each plot. Consequently, it could alter the simulated impact on hydrology and atmosphere. We could overcome this problem with the use of high spatial and temporal resolution imagery which allows observations of the land cover with a great precision. Particularly, it can monitor the actual phenological cycle of the crops (Etchanchu et al., 2017). It could thus offer precious

information to assess more accurately the crop water needs and consequently, the irrigation amounts and its effects on hydrometeorological fluxes. Some of the studies mentioned above already make use of satellite earth observation for irrigation estimations. But it mainly consists in classifying irrigated pixels, or correcting the previously mentioned irrigated areas maps with MODIS data (Ozdogan and Gutman, 2008; Evans and Zaitchik, 2008; Leng et al., 2013). The resolution of MODIS (500m) is unsuitable for the determination of irrigation rules in Western Europe because a MODIS pixel usually



contains several plots. This problem can be overcome by using high-resolution satellite data like LANDSAT, SPOT or the recent SENTINEL-2. Saadi et al. (2015) or Battude et al. (2017) already showed the potential of high-resolution products on determining the beginning and ending of the irrigation period in distributed semi-empirical crop models. However, these two studies do not make use of remote sensing to dynamically modulate the irrigation triggering with the actual phenology.

Hence, the present study suggests a method to use high resolution satellite products as input information for the proposed improved irrigation scheme. The balanced complexity between the schemes used in LSMs and agronomical models should allow taking full advantage of the increasing availability of high resolution data. A plot-by-plot validation is carried out on several plots in southwestern France during 2013 and 2014. We compare the simulated and observed irrigation volumes on each plot as well as the timing of the irrigations. Then, an evaluation of the impact on evapotranspiration is performed by

comparing with the eddy-covariance measurements of an instrumented site in the same region.

## 2 Model and data

### 2.1 SURFEX-ISBA model

#### 2.1.1 Model overview

EXternalized SURFace (SURFEX) is a modeling platform developed by the CNRM/Meteo-France. It aims at modeling the

water and energy transfers between the surface and the atmosphere (Masson et al., 2013). The version used in this study is the version 8.1. Within SURFEX, the model in charge of the simulation of the natural land surface is Interactions between Surface Biosphere Atmosphere (ISBA, Noilhan et Planton, 1989). ISBA uses meteorological forcing and land cover parameters for 19 plant functional types (PFT) to simulate coupled energy and water budgets. The computation cells can be composed of several patches with different PFT which are simulated separately. However, in this study, we used a single

PFT for each cell.

We used the standard version of ISBA in this study. The water transfer in the soil is modeled with a two layers force-restore approach (Deardorff, 1977). The first layer is the shallow topsoil layer. The second layer is the root-zone layer. This approach was integrated in ISBA by Mahfouf et Noilhan (1996). Water in the surface layer is drained and restored dynamically depending on the water content gradient between both layers. A gravitational drainage flux is simulated when

the soil water content of a layer exceeds the field capacity. Runoff only occurs when the surface layer is saturated. Concerning the energy fluxes, a unique energy budget is simulated for each vegetation type on the vegetation-soil layers composite by a single source scheme. A single surface temperature is used to solve the energy budget. This method is described by Noihlan and Planton (1989).

Concerning the irrigation, SURFEX-ISBA already incorporates a scheme that uses a 4 stages approach described by Voirin-

Morel (2003). Three parameters are required from the user: the dates of the first and last irrigation possible, and the water amount applied for an irrigation event. By default, the irrigation period spans from the 10[th] of May till the 1[st] of November





and the amount of each irrigation event is 30mm. Between these two dates, a soil moisture threshold is defined as in most LSMs (Sect. 1). The originality consists in a threshold that changes between the stages. Each stage is characterized by a different threshold. When the water content in the root-zone layer drops below this threshold, an irrigation event is triggered. The water depth is thus added to the precipitations of the current computation time-step. Then the model reaches the second

stage and uses another threshold for the next irrigation event. Irrigation cannot be triggered again within one day after the previous was applied. When the model reaches the fourth stage, the same threshold value is used for all the subsequent triggers. The threshold (named F2T hereafter) for each stage is defined as a fraction of the maximum available water content for root extraction (AWC$_{max}$) expressed in mm:

$$AWC_{max} = (W_{fc} - W_{wilt}) * D_{root} \quad (1)$$

where $W_{fc}$ is the volumetric soil water content at field capacity, $W_{wilt}$ the volumetric soil water content at wilting point and $D_{root}$ the rootdepth. To evaluate if an irrigation is needed, the model computes the current available water content (AWC) defined as follows:

$$AWC = (W - W_{wilt}) * D_{root} \quad (2)$$

where $W$ is the current volumetric soil water content. An irrigation event is triggered if AWC falls below a threshold defined

as a fraction of the maximum AWC: $AWC \leq (F2T * AWC_{max})/100$. The values are the following: F2T=70% for the first stage, F2T=55% for the second, F2T=40% for the third and F2T=25% for the last stage. The decreasing values for F2T were fixed to account for the root growth of the crop. Indeed, SURFEX-ISBA does not simulate the rooting depth evolution of the plants. It only uses a fixed root depth parameter which corresponds to the estimated rooting depth when the plant is mature. But the actual roots are only shallow in the first growth stages. Hence the actual AWC$_{max}$ is lower than the one estimated by

the model. As the water fluxes that enter and leave the soil-vegetation system are expressed in mm, the actual AWC/AWC$_{max}$ varies faster than the one simulated. For example, if the evapotranspiration flux is 5mm per day, then the filling rate of a soil where AWC$_{max}$=50 mm will decrease by 10% whereas a soil with AWC$_{max}$=100 mm will only lose 5%. Hence, farmers are particularly cautious at the beginning of the growth period to avoid water stress. To translate it in the model, the threshold for the first stage is high and decreases through the stages while the actual root depth reaches the value fixed in the model.

This irrigation scheme is initially designed exclusively for maize crops in Western Europe. Therefore, the only way to simulate irrigation in SURFEX-ISBA is to set the irrigated grid cells patches to the irrigated C4 crops PFT instead of their actual PFT (Masson & al., 2013). Consequently, all irrigated crops are simulated as irrigated maize. But many other crops than maize are actually irrigated, like soya or sunflower for example. They have very different land cover parameters and functional types. This scheme has several other limitations too. First, the fact that the irrigation is applied on a single

computation time step is not hydrologically realistic while computing with hourly or infra-hourly time steps as it could generate too much drainage or runoff. Then, the root growth is not simulated in diagnostic mode in ISBA. Hence the thresholds may not be adapted to this context. The 4 stages approach also does not take any information on the actual phenological cycle into account. It could lead to unrealistic irrigation timing. This approach also cannot take into account the crop sensitivity to water stress which depends on the growth stage. For example, the maize is particularly sensitive to water





stress during the flowering period whereas the sunflower has a high risk of disease if irrigated during the flowering period. These reasons lead us to develop a more generic and flexible irrigation scheme for SURFEX-ISBA.

## 2.1.2 Model development: improved irrigation scheme

To include a more realistic irrigation management, we have developed a new irrigation scheme for SURFEX-ISBA. The
goal was to develop a scheme which can be adapted to different crops and management types. In this scheme, seven parameters are needed: the dates of first and last irrigation possible, which depends on the growing period of the crop, the water amount of an irrigation event, the duration of each irrigation event, the minimal return interval between two successive irrigations, the irrigation type and the soil moisture threshold that triggers irrigation. The irrigation duration allows spreading the water depth within a certain time lapse as it is done in some studies (De Rosnay et al., 2003, Leng et al., 2017). For short
computation time-steps, like half-hourly or hourly time steps, it is assumed to be more realistic on a hydrological point of view than applying the water on a single time-step. Indeed, it generates less drainage and runoff. When estimating the available water resources, it also spreads the water uptake over a longer time span. It may result in a very different pressure on the water resources. The minimal return interval between two irrigation events is meant to represent the physical constraints on the water allocation, such as those defined by the network of canals or pipes. For example, pivot irrigation
systems cannot irrigate the entire field at the same time. It needs a certain time to irrigate again a specific part of the field after doing an entire turn to irrigate the other parts. In case of flood irrigation, this parameter translates the fact that the farmers must wait for their turn before opening the channels to their plots.

The irrigation type can be chosen amongst four different types: no irrigation, spray, drip or flood irrigation. This way, all vegetation patches can potentially be irrigated depending on the user's choice. This feature is new in SURFEX-ISBA since,
up to now, only C4 crops could be irrigated. Changing the irrigation type determines if the minimal return time must be applied or not. This parameter is applied only in spray and flood irrigation. During spray irrigation, the water is added to the precipitations at the top of the canopy, as in the previous scheme. For drip and flood irrigation, the water is applied at the top of the soil surface layer, below the canopy interception reservoir. In this study, we will only use the spray irrigation type. As in most of the models presented in section 1 (Lobell et al., 2009; Ozdogan et al., 2010, Battude et al., 2017), irrigation is
triggered when the water content in the root-zone soil layer drops below a threshold. This threshold is the most critical parameter in the scheme as it completely pilots the timing of the irrigation events. The novelty of our scheme is that the threshold can be forced spatially and temporally. Indeed, the threshold can differ from a computation cell to another and the user can force different values during time, on a daily, ten-days or monthly basis. Contrarily to of all the schemes mentioned in section 1, it allows the user to modulate the irrigation triggering rules based on its knowledge of the simulated area.
Particularly, it allows taking into account the actual phenology of the crop and irrigation management rules in the model. It thus offers the possibility to adapt the scheme to a wide variety of crops and irrigation management types while maintaining the possibility to use a constant threshold during time.



## 2.2 Datasets

### 2.2.1 Forcing data

The meteorological data used to force the model in this study are drawn from the Système d'Analyse Fournissant des Renseignements Atmosphériques à la Neige (SAFRAN, Durand et al. 1993) reanalysis data (Quintana et al., 2007). It
provides air temperature and specific humidity at 2 meters, air pressure, precipitations (in solid and liquid form), wind speed and direction, and solar radiation data. These data are provided at hourly time step on a 8 km resolution grid. These data were spatially linearly interpolated to match plot centroïds. For the soil parameters, we first used the Harmonized World Soil Database (FAO/IIASA/ISRIC/ISS-CAS/JRC, 2012), which gives the percentage of clay and sand at a 30 arc-second (~1 km) resolution. This database is used by default in SURFEX-ISBA. Soil parameters are then calculated by the model with
empirical pedotransfer functions (Noilhan and Lacarrère, 1995).

By default, SURFEX-ISBA uses the 1-km resolution ECOCLIMAP-II database (Faroux et al., 2013). This dataset describes land cover parameters as root depth, minimal stomatal resistance, albedo and LAI. The parameters evolving in time, like the LAI, are described with a climatology. As using a climatology is not sufficient to monitor the vegetation cycle and diagnose water needs, the LAI is derived from satellite imagery (cf. Sect. 2.2.1). But most of the other land surface parameters were
taken into the ECOCLIMAP-II database. As the simulated plots are exclusively maize plots in this study, only the root depth was forced, first with an empirical value of 0.9m, then with measured values.

### 2.2.2 Satellite Leaf Area Index

Chen and Black (1992) defined the Leaf Area Index (LAI) as "half of the total developed area of green (i.e. photosynthetic active) leaves per unit ground horizontal surface area". It traduces the surface of the plant which can contribute to the energy
and water exchange with the atmosphere. It is often used in land surface models as a proxy to represent the plant growth and thus, to scale the evapotranspiration of the plants. As the phenological cycle is determinant in the irrigation management, a LAI climatology is not sufficiently accurate in our study. Hence determining the LAI from high resolution satellite imagery allows taking the actual crop cycle into account in the model (Etchanchu et al., 2017).

Given the fact that the study focuses on years prior to Sentinel-2 launch (10 m resolution, 5 days revisit), we decided to
combine different satellite data to obtain a good revisit frequency. Thus we used data from Formosat-2, Landsat-8, SPOT-4 and SPOT-5. Formosat-2 is a Taiwanese satellite launched by the NSPO in 2004 and decommissioned in 2016. This satellite could acquire images on 4 spectral bands in the visible and near infra-red domain with a 8 m resolution. It did not make systematic acquisitions but work on tasking mode instead. The acquisitions had to be scheduled for specific areas. Its sensor detected the radiations within four frequency bands in blue, green, red and near-infrared. Landsat-8 is an American satellite
launched by the NASA in 2013. The optical sensor acquires in nine frequency bands : blue, green, red, near infra-red, two bands for mid infra-red, a panchromatic band and two bands respectively dedicated to aerosols and cirrus detection. Its spatial resolution is of 30m. Its revisit time is 16 days. SPOT-4 is a French satellite created in collaboration with Belgium





and Sweden. It was launched by the CNES in 1998 and finished its mission in 2013. Acquisitions were done on five frequency bands: green, red, near and mid infra-red, and a panchromatic band. It had a 20 m spatial resolution. Its temporal cycle was of 26 days. However, its adjustable mirrors allowed lateral acquisition on defined sites, giving it a 3 to 5 days revisit frequency. SPOT-5 was the successor of SPOT-4. It was launched in 2002 and ended its mission in 2015.

Acquisitions were done on the same bands as SPOT-4 but with a 10m spatial resolution. It also had adjustable mirrors to increase its revisit frequency to 3 to 5 days too.

All the measured reflectances have undergone geometric, atmospheric and radiometric corrections as well as a cloud detection filtering (Hagolle et al., 2008 and 2010).

To estimate the LAI, we used the neural network tool BV-NET, which inverts the PROSAIL radiative transfer model (Weiss

et al., 2002; Claverie, 2012). This neural network deduces a set of vegetation parameters (among them the LAI and the vegetation fraction cover, FCOVER) from the reflectance values for each pixel. It thus generates 8-m resolution LAI maps for each date and pixels without cloud obstruction. As BV-NET has a different calibration for each satellite sensor, there is a good consistency between the LAI produced with the data of the different sensors.

The time series of LAI maps were spatially averaged on the plots sampled during this study (Sect. 2.2.3 and 2.2.4). Maps

were then daily interpolated between available dates. As evolving land cover parameters are forced on a monthly basis in SURFEX-ISBA, the maps were finally temporally averaged to obtain monthly forcings of LAI for each plot of the study area.

All remote sensing and in situ measurements (Sect. 2.2.3) were collected as part of the Observatoire Spatial Régional (OSR) project for an agricultural area of southwestern France near Toulouse (Dejoux et al., 2012).

**2.2.3 Lamasquère flux site**

In order to evaluate the simulations, we used in-situ measurements performed at Lamasquère site (43°50'05" N, 01°24'19" E, Fig.1), which is included in the ICOS (Integrated Carbon Observing System) network. On this plot, an irrigated silage maize/winter wheat rotation is run and all the biomass of the plot is exported to feed the livestock of the adjoining farm. We only simulated years corresponding to silage

maize, e.g. 2006, 2008, 2010 and 2012.

At the plot, meteorological, soil and vegetation variables are monitored using standard equipments and set-ups. Half-hourly sensible heat flux and evapotranspiration are measured using an Eddy Covariance system, installed at 3.65 meters above the soil. The EC set-up consists in a fast open path Infrared Gas Analyzer (LiCor LI-7500, IRGA) and 3D-sonic anemometer components (Campbell, CSAT 3). The net

radiation and its components are measured with a CNR1 (Kipp & Zonen). The meteorological variables (and the dedicated sensors) are the wind speed (Windvane / prop Young), air temperature and humidity





(HMP35, Vaisala). The soil profile probes for water content measurements (CS616, Campbell Scientific) are buried at depths of 5 cm, 10 cm, 30 cm, and 60 cm and also 100cm underground. All turbulent fluxes computation like evapotranspiration (ETR), latent heat (LE) and sensible heat (H) fluxes are calculated at a half-hourly time step according to CarboEurope-IP recommendation (Aubinet et al., 1999, Béziat et al, 2009). The flux computation includes spectral frequency and Webb corrections (Moore, 1986, Webb et al., 1980) and various filters and statistical tests are performed on the data fluxes to remove out of range and non-stationary data (Papale et al., 2006; Reichstein et al., 2005). The spatial representativeness (footprint) of the fluxes is also taken into account and if the calculated fetch including 90 % of the flux (Kljun et al., 2004) model for each half-hourly EC flux value (*F-90*) was higher than the distance between the eddy-covariance tower and the edge of the plot in the main wind direction, fluxes were discarded. Eventually, the data were gapfilled to generate a full yearly database and depending on the duration of missing data, either the linear regression method or the mean diurnal variation was applied according to Beziat & al. (2009).

Technical itinerary has also been given by the farmer. It contains information about the plot management and agricultural practices such as the sewing and harvest dates, the tillage characteristics, the fertilization, the herbicide spray and the dates and depths of irrigation events.

### 2.2.4 CACG irrigation data

The Compagnie d'Aménagement des Coteaux de Gascogne (CACG) is in charge of the management of the Neste channel which supplies 17 rivers in southwestern France. The targeted sector is mainly agricultural. Hence, the CACG is also in charge of advising farmers on irrigation practice in order to optimize the water resources availability. To forecast the availability of water resources at seasonal scales, the CACG monitors several irrigated plots every year. For each of these plots, the log containing the field operations has been collected. Thus, the irrigation dates and amounts are gathered. In this study, we used the data from the years 2013 and 2014. These data were collected as part of the MAISEO project (http://www.pole-eau.com/Les-Projets/Projets-innovation-finances/Maiseo). Data were available on 23 plots in 2013 and 13 in 2014 spread across the Haute-Garonne and Gers French departments (Fig. 1). For some of these plots (10 in 2013, 11 in 2014), soil measurements have been performed. Particularly, an estimation of the root influence depth is provided.





## 3 Methods

### 3.1 Determination of the irrigation parameter

As this study focuses on maize crops in southwestern France, we determined simple decision rules based on expertise over this area. To determine the irrigation period, we used a LAI threshold. For the maize in South-Western France, the irrigation

should not begin before the 8-10 leaves growing stage according to agencies (CACG or Arvalis ) in charge of advising the farmers. Measurements showed that this stage corresponds to a LAI value of nearly 0.4 m$^2$ m$^{-2}$. The irrigation period thus begins when the LAI increases above 0.4. A minimal date, the 1$^{st}$ of May, has to be used to filter intermediate winter crops. Note that this date is only adapted to crop practices in France. As the LAI will not decrease under 0.4 before harvest, we also used this threshold to detect the harvest. The irrigation period is assumed to end 45 days before the harvest. This 45 days

period corresponds to the time needed for the grain to dry between last irrigation and harvest. This value is also used by Battude et al. (2017). For the irrigation amount and minimal return time, the values were based on technical documentation from these agencies too. The irrigation amount was set to 30mm per event and the minimal return time to 6 days based on the observed irrigations and the capacities of typical irrigation materials used on maize crops in South-Western France. The duration of an irrigation event depends on the irrigation material nominal discharge but each part of a plot is generally

irrigated during about 8 hours. We therefore used 8 hours as the application time for irrigation.

At last, we used the remotely sensed LAI to pilot the root-zone soil moisture thresholds. This represents the major improvement of our study. Indeed, in southwestern France, the growing stage of the maize crops modulates the irrigation practice. In the early stages of the growth, the root depth is low and so is AWC$_{max}$. Hence, as explained in section 2.1.1, the filling rate of the maximal available water content (AWC/AWC$_{max}$) varies faster. The farmers are thus advised to anticipate

for dry periods between two irrigation events in the early stages. As SURFEX-ISBA does not simulate the root growth, the maximal AWC remains fixed to the one when the roots are fully developed. Therefore, to simulate the farmer practices, the first threshold of the irrigation period must be higher than what is used for a fully developed root zone. Following the advice of CACG and Arvalis, we used a first threshold of F2T$_{max}$=80% of AWC$_{max}$. As the actual root depth increases during the growth, the threshold decreases. To represent this effect, we applied the LAI dynamic to the decrease with the following

relationship:

$$F2T_i = F2T_{max} + \frac{(LAI_i - LAI_{max})}{(LAI_{max} - LAI_{min})} * (F2T_{max} - F2T_{flw}) \ (3)$$

where F2T$_i$ is the AWC threshold for the month *i*, given in % of AWC$_{max}$, F2T$_{flw}$ the AWC threshold at the flowering stage, LAI$_i$ the mean LAI of the plot for the month *i* and LAI$_{min}$ and LAI$_{max}$ respectively the minimal and maximal monthly averaged LAI of the current year. The maize reaches its maximal LAI when the flowering period begins. In this period, the

crop is very sensible to water stress as it may impact the yield. As a consequence, farmers are advised to maintain AWC above 60% of AWC$_{max}$ as a safety margin. Hence, F2T$_{flw}$=60%. Once the flowering period is over, as the plant goes through maturation, this sensitivity to water stress decreases. The objective for the farmers is to let the crop exploit the water in the soil without limiting the biomass production. They thus try to use what is defined by Allen et al. (1998) as the readily





accessible water content. The value proposed by Allen et al. (1998) is 55% of $AWC_{max}$ for the maize crops. It implies farmers must maintain AWC above 45% of $AWC_{max}$ until the end of the crop cycle. This threshold value is the one used in the SAFYE model (Battude et al., 2017). In fact, the easily accessible water content depends on the plant but also on the soil texture. Following the expertise of the CACG, based on their measurements, we decided to set the minimal threshold to

$F2T_{min}$=40%. The thresholds between flowering and harvest are then defined by the following relationship:

$$F2T_i = F2T_{min} + \frac{(LAI_i - LAI_{min})}{(LAI_{max} - LAI_{min})} * (F2T_{flw} - F2T_{min}) \quad (4)$$

This way, the threshold decrease is entirely timed by the phenology, contrarily to the actual irrigation scheme (Fig. 2). However, the limit values $F2T_{max}$, $F2T_{flw}$ and $F2T_{min}$ are based on expertise of the maize crops management in South-Western France. For applications on other region or other crops, with different irrigation management, the parameters must

be adapted.

### 3.2 Numerical experiments

In this study, four numerical experiments were done.

- STD which uses the standard irrigation scheme of SURFEX-ISBA with standard irrigation parameters. It is the reference simulation.

- STD-SAT which uses the standard irrigation scheme of SURFEX-ISBA with an irrigation period derived from the remote sensing LAI (Sect. 3.1). This experiment aims at showing an eventual contribution of using remote sensing to modulate only the irrigation period.

- FIXE which uses the new irrigation scheme developed in this study with the decision rules described in section 3.1 but with F2T being kept constant to 45%, as used by Saadi et al. (2015) and Battude et al. (2017).

- VARI which uses the new irrigation scheme with all the decision rules described in section 3.1.

A summary of the numerical experiments and their parameters is available in table 1.

We also performed these 4 experiments on the Lamasquère plot and on the CACG plots separately. Note that for the Lamasquère plot, the irrigation event duration and maximum irrigation frequency are different from the table, as described in section 3.1. Hence, the minimal return time for irrigation could be set to 5 days and the irrigation duration to 5h, according to

the characteristics of the irrigation material. On this plot, the crop cultivated is silage maize which is harvested while still green. So the grain does not have to dry before harvest and can be irrigated until the harvest. Consequently, we also changed the ending of the irrigation period criterion. It was directly when the LAI drops under 0.4 m² m⁻² instead of using a 45 days drying period. However, as silage maize is hard to segregate from standard maize in classification algorithms, this criterion must only be used on well documented plots.

First, we fix the rooting depth to 90 cm based on the measurements done on the Lamasquère site. Then, Section 4.1.2 presents the impact of changing the rooting depth, and consequently the estimation of $AWC_{max}$, on the CACG plots. In this section, we use the root depths measured by the CACG on some of the plots.



For the Lamasquère plot, soil moisture initialization is directly taken from in-situ measurements. As we do not have this kind of measurements on the CACG plots, the initialization for all the experiments was determined with a spin-up run based on the STD experiment as it is the reference simulation.

## 4 Results

### 4.1 Irrigation needs estimation on CACG plots

#### 4.1.1 CACG plots: irrigation volumes estimation

To evaluate the new irrigation scheme, we first compare the annual irrigation volume on each CACG plot for the different options and observations. The results are shown in figure 3. The corresponding scores are summarized in table 2.

For the year 2013 (Fig. 3.a), the VARI experiment performs the best. The bias is close to null and the RMSE is around the volume of an irrigation event. The three other experiments underestimate the volume of irrigation compared to the observations, by omitting an average of two irrigation events during the season. Looking at the timing of the first event (Fig. 4.a, table 3), all the experiments performs pretty well with a few days bias and a RMSE around a week. FIXE overestimates the first irrigation date due to its tendency to underestimate the irrigation needs. In STD and STD-SAT experiments, this effect is overriden by the decreasing threshold, which means that the irrigation needs for the first event are well estimated. Concerning the last irrigation timing (Fig. 4.b, table 4), there is little difference between experiments. STD and STD-SAT slightly underestimate this date due to the very low threshold in later stages. However, all experiments also perform well with a low bias and a RMSE around a week and a half. There are few differences between STD and STD-SAT for the irrigation amount as well as for the dates of first and last irrigation. This is because in the standard irrigation scheme, the irrigation triggering thresholds are the same, independently from the beginning of the irrigation season. Hence, if the first irrigation happens after the beginning of the season in both STD and STD-SAT, then the irrigation amounts and timings will be the same. This is also true for the end of the irrigation period.

The summer of 2014 is more humid than the 2013 one with precipitation up to 214 mm during the June-July-August period in 2014 against 148 mm in 2013 (171 mm in average on the 1981-2010 period). This explains why the observed and simulated irrigation amounts are much lower (Fig. 3.b). Table 2 shows that STD experiment performs best in 2014, with performances similar to VARI in 2013. It is followed by VARI and STD-SAT. These two experiments still show good RMSE, around one and a half volume of an irrigation event. The difference between STD and STD-SAT is due to the fact that the end of the irrigation season estimated by satellite occurs sooner than the default date set in SURFEX-ISBA. FIXE shows the worst performances this year. However, it is interesting to note that the scores for the new irrigation scheme (FIXE and VARI) are quite consistent through the years. Concerning the dates of first irrigation, STD and STD-SAT perform quite well, with scores close to those obtained in 2013. On the contrary, FIXE deeply overestimates the first irrigation date. VARI shows a good bias but is way more uncertain than STD and STD-SAT, with a RMSE of 26 days. As




for the last irrigation dates, STD and VARI strongly overestimate the last irrigation date with better performances of STD. On the contrary, STD-SAT and FIXE largely underestimate this date. The more uncertain estimation of the first and last irrigation timing will be discussed in section 5.

### 4.1.2 Uncertainty on the maximal available water content estimation

The irrigation triggering in both irrigation schemes mostly relies on the threshold values. As these thresholds are expressed in percentage of $AWC_{max}$, which depends on the prescribed soil hydrodynamic parameters and root depth, its estimation appears to be crucial. This section aims at evaluating the impact of the uncertainty from $AWC_{max}$ on the simulated irrigation volumes. In order to do so, we use the measured root depths of the CACG. For the plots where no measurement is available, the root depth is kept to 90 cm. As the irrigation volumes are small in 2014, we decided to focus on 2013 in this analysis.

The scores on annual volumes and first and last irrigation timing are showed in Table 5. The difference with the scores calculated with the 90cm root depth is also shown (see parenthesis). For all the experiments the irrigation volume estimated is increased by around 20 mm compared to the simulations with the fixed root depth. It means that for a large fraction of the plots, a supplementary irrigation event was added. The RMSE is not impacted that much for STD, STD-SAT and FIXE experiments. On the contrary, the VARI experiment becomes more uncertain with a 72 mm RMSE, 34 mm larger than the

one obtained with a fixed root depth. The timing of first and last irrigations remains well estimated for all experiments, with small differences between the fixed and measured root depth simulations. To further understand these results, we decided to analyze the differences on each plot. Figures 5, 6.a and 6.b respectively show the error on the annual irrigation volume, the error on the first irrigation date and on the last irrigation date depending on the prescribed root depth. It may first be noticed that the measured root depths are mostly way smaller than 90 cm. Thus $AWC_{max}$ is lower for these plots. Therefore, for the

same atmospheric water demand, and thus the same evapotranspiration flux, the filling rate of $AWC_{max}$ drops faster. It leads to supplementary triggers, explaining the 20 mm increase in the bias for the annual volumes estimated. A general trend appears with a clear overestimation of the simulated irrigation volumes when the root depth decreases below 90cm. The same trend is visible on the dates of first and last irrigation with an underestimation of the first irrigation date and an overestimation of the last irrigation date. The VARI experiment seems to be the most affected, explaining the big difference

in RMSE between the fixed and measured root depth simulations. This point will be discussed in section 5.

### 4.2 Impact on evapotranspiration: case of the Lamasquère site

In order to estimate the impact on the hydrometeorological fluxes, the four experiments described in section 3.2 were also conducted on the Lamasquère flux site. An evaluation of the irrigation volumes is first performed for each year with irrigated maize crop. Then we compare the simulated evapotranspiration and sensible heat fluxes with eddy covariance measurements

onsite.



### 4.2.1 Irrigation volumes estimation

Figure 7 shows the annual irrigation volumes for each year at Lamasquère. The scores on irrigation volumes and dates are summarized in table 6. STD, STD-SAT and FIXE show similar scores with around -30 mm of bias and 40 mm of RMSE. It means that these three experiments underestimate the crops actual irrigation by more than an irrigation event. As on the CACG plot in 2013, the VARI experiment irrigation volume estimations are better. The bias is near zero and the 22mm RMSE is under the volume of an irrigation event. However, all the experiments show pretty good results. As on the CACG plots, we have also checked the dates of first and last irrigation (Table 6, Fig. 8.a and 8.b). FIXE substantially overestimates the first irrigation date compared to the observed ones (Fig. 8.a). The three other experiments have similar performances as on the CACG plots in 2013 with a bias around a week and a RMSE of a week and a half. The estimation of the last irrigation timing is accurate for all the experiments (Fig. 8.b), showing performances equivalent to the estimations on the CACG plots in 2013 too.

### 4.2.2 Impact on evapotranspiration

Figure 9 shows the simulated monthly evapotranspiration every year for each experiment compared to the measured ones. Scores are summarized in table 7. The monthly evapotranspiration is well represented by all the experiments. The differences are low between experiments. All the experiments have the same RMSE on the daily evapotranspiration. However, when looking at the cumulated evapotranspiration over the year, VARI performs the best with a -14 mm yr$^{-1}$ bias whereas the three other experiments show a bias of around -30 mm yr$^{-1}$. This result is coherent with the fact the 3 other experiments tends to underestimate the annual irrigation volume. It also means that a 30 mm difference in annual irrigation volume does not lead to the same difference in evapotranspiration. This point will be discussed in section 5.

## 5 Discussion

### 5.1 Estimation of the irrigation volumes and timing

We will first discuss the results related to the CACG plots. Concerning the annual irrigation volumes estimation, the performances for the new scheme, in the FIXE and VARI experiments, are quite consistent between years (Table 2). Hence, the new scheme appears more stable than the standard irrigation scheme of SURFEX. Indeed STD and STD-SAT have very different performances in 2013 and 2014. The difference between FIXE and VARI shows the limits of directly applying the formalism of agro-hydrological models like SAFYE (Battude et al., 2017) in a Land-Surface Model. Indeed, the FIXE experiment, which is a direct application of the SAFYE formalism with the same parameter values, cannot catch up with such performances on the estimation of annual irrigation volumes, contrarily to the VARI experiment. It shows that the method used in VARI to determine the irrigation triggering thresholds fits better to the average irrigation management of maize crops in our study area. For both years, FIXE underestimates the irrigation volumes. Indeed the threshold at 45% of





AWC$_{max}$ represents an "optimal" use of the water without taking any safety margin. Consequently, the soil moisture may drop below the threshold during the minimal return interval between two irrigation events. It may lead to drought stress and affect the yields. VARI takes into account the fact that farmers will take safety margins and simulates it. The STD and STD-SAT have poor performances in 2013 because of a strong underestimation of the crop water needs. It is due to the threshold

used for the last stage which is too low. It makes it hard for the standard irrigation scheme to simulate more than three irrigation events. The problem does not occur in 2014 which had a wet summer. The observed irrigation volumes are lower and easier to catch with the standard module.

The irrigation period is well represented in 2013 as shown in table 3 and 4, and figure 4. It means that the STD and STD-SAT experiments underestimate the irrigations in the middle of the season. The timing of the soil moisture threshold seems

responsible for this weakness because its decrease is not constrained by phenology. For 2014 (Fig. 4.c and 4.d), the timing of irrigation is more uncertain for all the experiments, especially when estimating the last irrigation event. As observed, the irrigation volumes are low in 2014, as well as the number of actual irrigation events. Hence, they are concentrated in a small period which corresponds to the dryer period of the season. In this wet context, farmers' decisions seem to be less correlated to the soil water content. The model is thus likely to simulate out of phase irrigation events, as shown in tables 3 and 4 for

2014. However, given the low volumes observed in 2014, it should not significantly affect the hydrology.

The results on irrigation estimation for Lamasquère (Sect 4.2.1) confirm that the VARI experiment fits best the irrigation practices of the area. However, this result may be modulated by the fact that Lamasquère is an experimental plot of the agricultural engineering school of Purpan. As scientific experiments may be performed on this plot, the irrigation management does not necessarily correspond to what usually happens in South-Western France.

**5.2 Sensitivity to the maximal available water content estimation**

Section 4.1.2 shows that the estimation of AWC$_{max}$ can strongly affect the results. Especially, for all experiments, the error on the annual irrigation volume estimation increases with decreasing root depth. This result may be explained by two points. The first is that when AWC$_{max}$ is low, its filling rate varies faster. The model is thus very sensitive to the formalisms in charge of drying the soil layers like drainage and evapotranspiration. If one of these formalisms generates excessive drying

of the soil layers, the scheme will simulate way too much irrigation events, as shown in figure 5. It is especially true when the thresholds are high, explaining why the VARI experiment is more affected than the other ones. This reason should push the users to overestimate a little bit the rooting depths rather than underestimating it as it should generate smaller errors. The second plausible cause is to assume that the farmers do not adapt their practices to their soil. In fact, most of the CACG plots have low soil depths. It thus limits the root development of the maize crop, contrarily to Lamasquère. In such conditions, the

farmers are usually advised by technical institutes to diminish the irrigation amount of each event and increase the frequency. For example, they should adapt their irrigation material to irrigate 20 mm with a 4-days return period, instead of 30 mm each 6 days. But in the observed irrigation series for the CACG plots, the mean irrigation volume for each irrigation event is very close to 30 mm with a small dispersion. It could mean that the farmers are not fully aware that an impervious



soil layer, related to the geomorphology of the landscape in the Gers department, limits the root development in their plots. They thus follow   advices of technical institutes based on a soil without constrain on rooting depth. Hence, it may explain why VARI performs way better than the other experiments with a fixed 90 cm root depth (Sect. 4.1.1 and 4.2.1).

## 5.3 Impact on evapotranspiration

When looking at the impact on evapotranspiration (Sect. 4.2.2), the differences between experiments and observations are close to the measurement precision. Uncertainty mainly comes from the closure of the energy budget in the measurements, which was around 90% for Lamasquère in 2006 (Beziat, 2009). However, there is a slight impact on yearly cumulated evapotranspiration. VARI is the experiment that most reduces the bias, from around -30 mm yr$^{-1}$ to -14 mm yr$^{-1}$. As all the experiments are designed to avoid water stress on the crops, evapotranspiration is pretty close to the potential one.
Differences are essentially due to differences in soil evaporation. A 15 mm yr$^{-1}$ difference might seem a low value but this difference is concentrated in periods of high demand and low availability of the water. It may thus influence decisions concerning the water uptake from the river. It is also interesting to note that the 30 mm yr$^{-1}$ difference between VARI and the three other experiments on the Lamasquère plot (table 6) lead to only 15 mm of difference on evapotranspiration. It means that only 50% of the supplementary water added by VARI contributes to evapotranspiration on a yearly basis, giving an
estimation of the efficiency level of the spray irrigation.

## 5.4 Limitations and perspectives

These results should however be taken carefully due to the uncertainties in the different processes involved. First of all, various uncertainties of the satellite acquisition must be taken into account. The acquisition performances of all the satellites used in this study are similar to those of Sentinel-2 (Koetz et al., 2017). Before averaging the LAI at the field scale, we
eroded the plots' geometries by the width of a pixel to avoid border effect and geo-location uncertainty. But few CACG plots were smaller than a pixel after this treatment. This problem could affect the representativeness of the LAI on these plots. The radiometric uncertainties on each band are acceptable, as well as the Signal-to-Noise ratio.  The LAI estimation is also well known to saturate (Veloso et al., 2012). But above a certain threshold, the difference between satellite-estimated and real LAI does not affect significantly the evapotranspiration flux (Etchanchu et al., 2017).  The estimated LAI may also differ
between sensors. But the calibration of BVNET was adapted to each sensor, meaning that the differences between sensors may be smoothed by BVNET. The main uncertainty remains in the estimation of the beginning and ending of the irrigation season. Indeed, these are based on LAI threshold. So a missing date due to excessive cloud coverage around these periods results in an inaccurate tracking of the phenology. This effect is attenuated by the multi-sensor approach used in this study. Indeed, it allows increasing the number of clear-sky acquisitions.
The main limitations still lie in the application of the method presented at larger scale. First of all, the determination of the irrigated surfaces remains a challenge for large areas. The French National Centre for Space Studies (CNES) is actually coordinating the research effort in France.  The main point is to use high resolution remote sensing products to detect



irrigated plots over an entire river basin. The second limitation for regional applications is the fact that the irrigation practices strongly vary from a region to another and between crops. Although remote sensing products can provide information about the irrigation period delimitation and a way to adapt the irrigation decision rules to the actual phenology, knowledge about the common irrigation practices for each crop in the area remains essential. Without any upstream

expertise, the best practice would be to collect irrigation measurements representative of each crop over the entire area, and then, to calibrate the parameters to fit the local irrigation practices. At last, evaluating the impact of irrigation practices on water resources requires coupling SURFEX-ISBA with a hydrological and a hydro-geological model. Only then we could estimate the impact on the river discharge and on the water table level evolution.

**Conclusion**

This study presents an irrigation scheme with a balanced complexity lying between the complexity level of most actual land surface models and advanced agronomical models. The increased flexibility allows this new scheme to accommodate a wide variety of irrigation practices and crop types, since it can use information on actual phenology, while maintaining the possibility to set simple rules. However, increasing the complexity also means that the model requires more input information. Hence this study makes use of high resolution remote sensing products to determine part of this information.

The STD experiment, using the actual irrigation scheme of SURFEX-ISBA, was taken as a reference. The STD-SAT experiment was designed to evaluate the sole contribution of remotely sensed irrigation period. The FIXE experiment was designed to test a common parameterization for the irrigation scheme based on the SAFYE model. At last, the VARI experiment was designed to fully exploit the potential of the new scheme for using additional information like remote sensing products. The experiments were conducted on several maize plots monitored by the CACG for two years (2013 and

2014) as well as on the instrumented site of Lamasquère. The results show that the VARI experiment succeeds in using this additional information. Indeed, it generally gives better estimation of the irrigation volumes as well as the irrigation events timing. This conclusion is less true than for a wet year (2014) with small irrigation volumes concentrated on a small time period. The sensitivity analysis also showed that the VARI experiment seems a little more sensitive to the estimation of the maximal available water content due to its higher triggering thresholds. This analysis also points out the subjectivity of the

farmer's decisions and the difficulty to model anthropogenic effects. Therefore, for regional applications, the parameters of the irrigation should be calibrated for each crop type to fit the local irrigation practices. This adaptability makes our irrigation scheme a strong tool to assess the impact of different irrigation practices and their evolutions on the water resources.





**Code and data availability**

The version of SURFEX used in this study is the v8.1. The code is available here: http://www.umr-cnrm.fr/surfex/data/BROWSER/doc_surf81/index.html.

For the LAI, evapotranspiration or irrigation data, please ask to the corresponding author.

**Author contribution:**

J. Etchanchu formatted the data, performed the simulations, analyzed the results and wrote most of the current paper.

V. Rivalland and G.Boulet helped at designing the experiments and analyzing the results.

S.Faroux helped at integrating the new irrigation scheme in the current version of SURFEX-ISBA.

A. Brut wrote the section 2.2.3 about the instrumentation of the two stations and helped analyzing the results on evapotranspiration.

All the authors revised the paper.

**Acknowledgments:**

We thank Marjorie Battude for her work on the irrigation estimation which constitutes the basis of this study. We also thank Jean-François Dejoux for his expertise on the maize crops in order to define the irrigation rules as realistic as possible. Thanks to Valérie Démarrez for her advices concerning the CACG plots and the irrigation processes.

We thank Ludovic Lhuissier, Mamadou Diarra and Jean-Jacques Webber, from the CACG, for allowing us to use some of their data and for their expertise on the irrigation management of the maize crops.

We thank all the people of the OSR who contributed to the collect and the process of data. OSR facilities and staff are funded and supported by the Observatory Midi-Pyrenean, the University Paul Sabatier, Toulouse, France, CNRS (Centre National de la Recherche Scientifique), CNES (Centre National d'Etude Spatial) and IRD (Institut de Recherche pour le Développement). We especially thank the European Research Infrastructure Consortium Integrated Carbon Observation System (ERIC ICOS) and ICOS-France for funding facilities and staff working on Lamasquère sites.

We also thank the BAG'AGES project for its financial support.

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





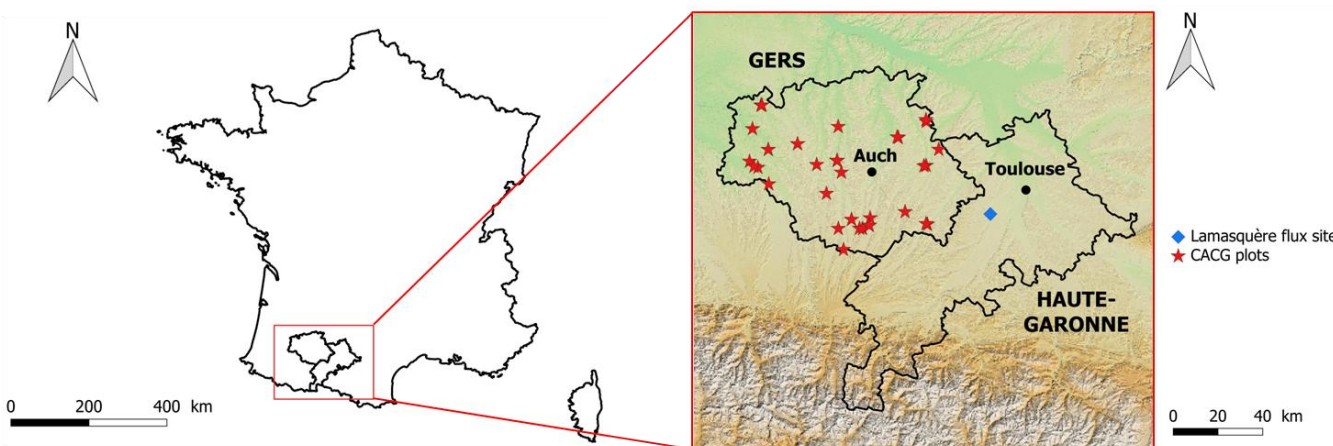

**Figure 1: Study area**

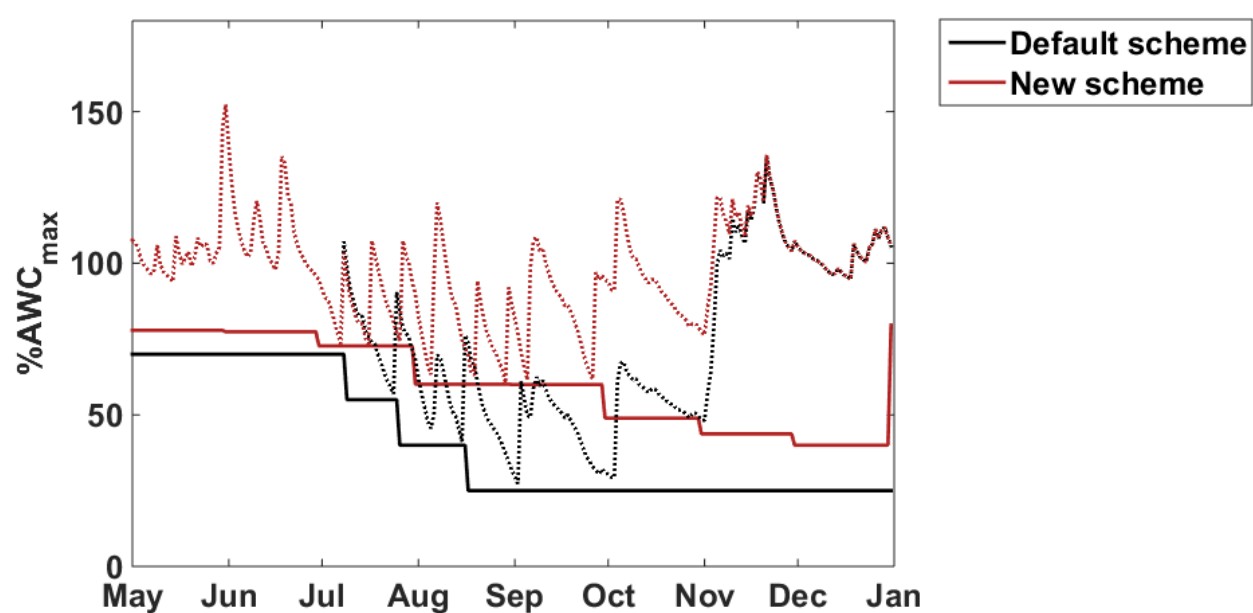

**Figure 2: Irrigation triggering thresholds for the default and new irrigation schemes ISBA for the CACG plot n°9 in 2013. The dashed lines are the simulated soil moisture in the root zone with both schemes.**



(a)

(b)

**Figure 3: Annual irrigation volumes for the CACG plots in 2013 (a) and 2014 (b)**







**Figure 4: Observed and simulated first (a, c) and last (b, d) irrigation event dates for the CACG plots in 2013 (a, b) and 2014 (c, d)**



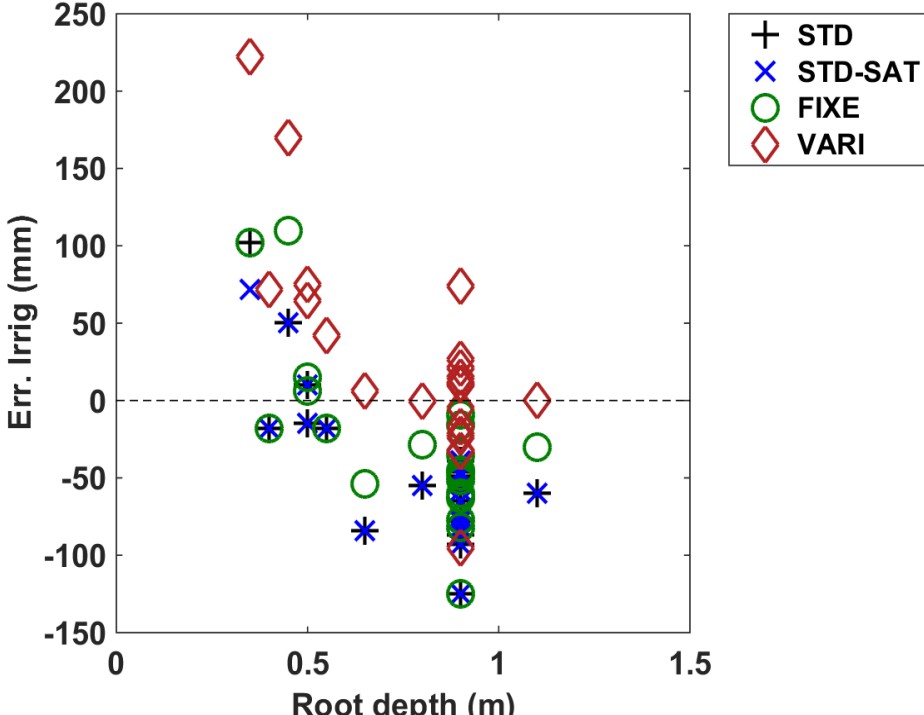

**Figure 5: Error on the estimation of annual irrigation volumes for the CACG plots in 2013 as function of the rooting depth**

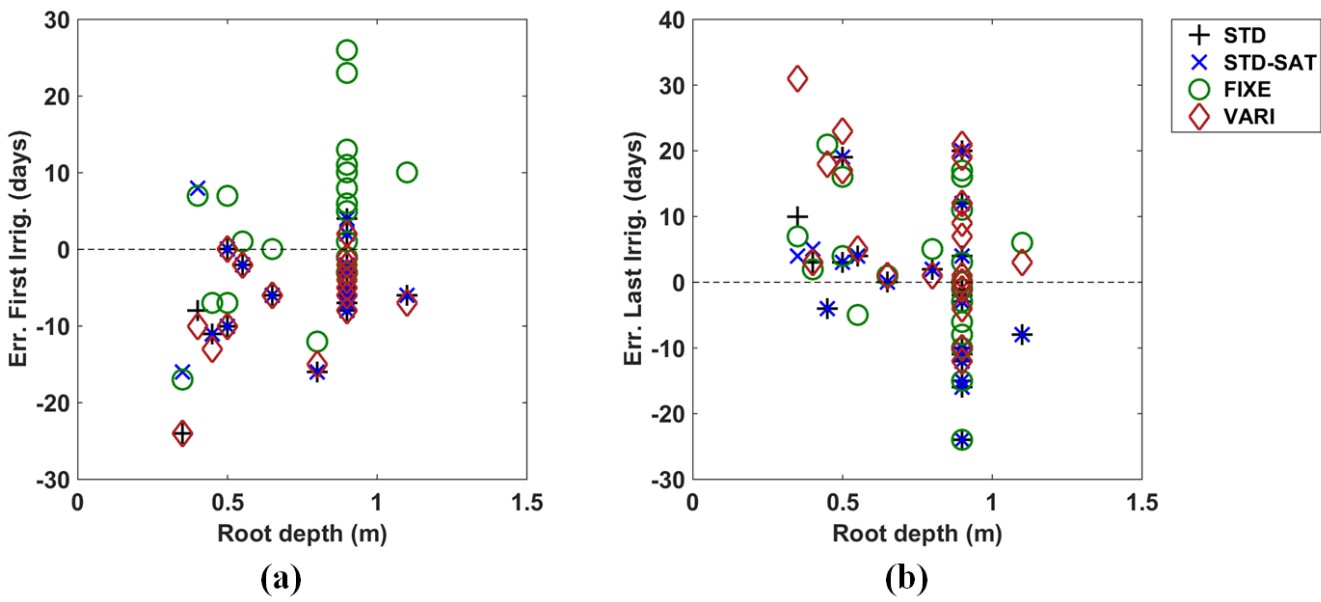

5    **Figure 6: Error on the first (a) and last (b) irrigation event dates for the CACG plots in 2013 as function of the rooting depth**



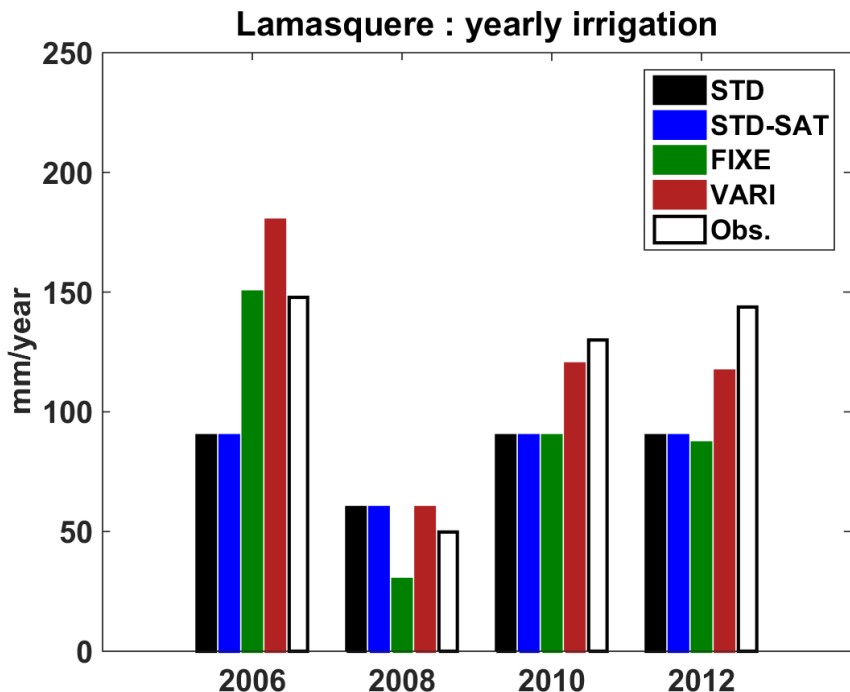

**Figure 7: Annual irrigation volumes for the Lamasquère site**

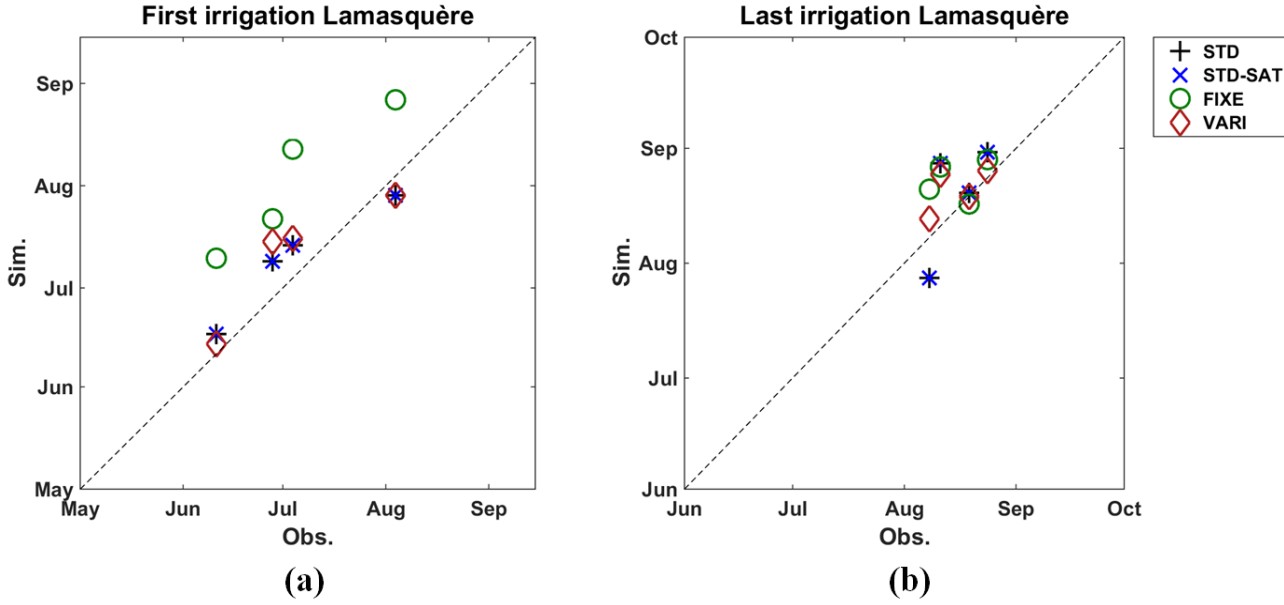

**Figure 8: Observed and simulated first (a) and last (b) dates of irrigation events for each year for the Lamasquère site**





**Figure 9: Monthly evapotranspiration on the Lamasquère site in 2006 (a), 2008 (b), 2010 (c) and 2012 (d)**



| Experiment | Irrigation scheme (standard or new) | Irrigation period | Irrigation amount (mm) | Irrigation event duration (h) | Irrigation maximal frequency (day) | Irrigation triggering soil moisture thresholds (% max AWC) |
|---|---|---|---|---|---|---|
| STD | Standard | ECOCLIMAP | 30 | 0.5 | 1 | Decreasing (stages): 70-55-40-25 |
| STD-SAT | Standard | Satellite | 30 | 0.5 | 1 | Decreasing (stages): 70-55-40-25 |
| FIXE | New | Satellite | 30 | 8 | 6 | 45 |
| VARI | New | Satellite | 30 | 8 | 6 | Decreasing (LAI): 80~60~40 |

**Table 1: Summary of the numerical experiments**

| Exp. | 2013 | | 2014 | |
|---|---|---|---|---|
| | Bias (mm yr$^{-1}$) | RMSE (mm yr$^{-1}$) | Bias (mm yr$^{-1}$) | RMSE (mm yr$^{-1}$) |
| STD | -61 | 68 | **-5** | **37** |
| STD-SAT | -64 | 71 | -32 | 38 |
| FIXE | -47 | 58 | -52 | 66 |
| VARI | **2** | **38** | 22 | 49 |

5  **Table 2: Scores on cumulated irrigation volumes on the CACG plots in 2013 and 2014**





| Experiment | 2013 | | 2014 | |
|---|---|---|---|---|
| | RMSE (days) | Bias (days) | RMSE (days) | Bias (days) |
| STD | **7** | **-4** | 14 | 0 |
| STD-SAT | **7** | **-4** | **10** | **3** |
| FIXE | 12 | 8 | 20 | 16 |
| VARI | **7** | **-5** | 23 | -3 |

**Table 3: Scores on the estimation of the first irrigation event dates on CACG plots in 2013 and 2014**

| Experiment | 2013 | | 2014 | |
|---|---|---|---|---|
| | RMSE (days) | Bias (days) | RMSE (days) | Bias (days) |
| STD | 11 | -3 | **25** | **20** |
| STD-SAT | 11 | -5 | 39 | -34 |
| FIXE | **10** | **0** | 36 | -23 |
| VARI | 10 | 5 | 39 | 34 |

5    **Table 4: Scores on the estimation of the last irrigation event dates on CACG plots in 2013 and 2014**

| Exp. | Annual irrigation volume | | First irrigation date | | Last irrigation date | |
|---|---|---|---|---|---|---|
| | Bias (mm yr⁻¹) | RMSE (mm yr⁻¹) | RMSE (days) | Bias (days) | RMSE (days) | Bias (days) |
| | Bias (mm $yr^{-1}$) | RMSE (mm $yr^{-1}$) | RMSE (days) | Bias (days) | RMSE (days) | Bias (days) |
| STD | -44 (+17) | 66 (-2) | **9 (+2)** | **-6 (-2)** | **11 (+0)** | **-2 (+1)** |
| STD-SAT | -47 (+17) | 65 (-6) | **8 (+1)** | **-5 (-1)** | **11 (+0)** | **-3 (+2)** |
| FIXE | **-29 (+18)** | **60 (2)** | 11 (-1) | 4 (-4) | **11 (+1)** | 2 (+2) |
| VARI | **26 (+24)** | **72 (+34)** | **8 (+1)** | **-6 (-1)** | 12 (+2) | 6 (+1) |

**Table 5: Scores on the annual irrigation volume and the first and last irrigation dates on CACG plots in 2013 with the measured root depths. Within the parenthesis is the difference with the score obtained while simulating with a root depth fixed to 90cm**
10    **(Table2).**





| Exp. | Annual irrigation volume | | First irrigation date | | Last irrigation date | |
|---|---|---|---|---|---|---|
| | Bias (mm yr$^{-1}$) | RMSE (mm yr$^{-1}$) | RMSE (days) | Bias (days) | RMSE (days) | Bias (days) |
| STD | -35 | 45 | **9** | **5** | 11 | 4 |
| STD-SAT | -35 | 45 | **9** | **5** | 11 | 4 |
| FIXE | -29 | 36 | 29 | 29 | 11 | 8 |
| VARI | **1** | **22** | 11 | 7 | **8** | **5** |

**Table 6:** **Interannual scores on the annual irrigation volume and the first and last irrigation dates on Lamasquère**

| Exp. | Bias (mm yr$^{-1}$) | RMSE (mm d$^{-1}$) |
|---|---|---|
| STD | -31 | 0.93 |
| STD-SAT | -31 | 0.93 |
| FIXE | -31 | 0.93 |
| VARI | **-14** | **0.94** |

**Table 7:** **Interannual scores on evapotranspiration on Lamasquère**