# Peer review of "On the use of high resolution satellite imagery to estimate irrigation volumes and its impact in land surface modeling"

_Hydrology and Earth System Sciences, 2019_

## Referee Comment (RC1) · Anonymous Referee #1 · 13 May 2019

The paper proposes the development of an irrigation module in LSM that better reflects the reality of decisions that depend on the crop, its phenological stage and irrigation techniques. The study found that current models are either too simplistic with the impossibility of representing real practices or too complex with the impossibility of providing large-scale information on the parameters needed to characterize the irrigation schedule. The idea is to improve the irrigation scheme present in the ISBA model by using remote sensing imagery to characterize phenology (actually 3 stages: the 8-leaf stage with an LAI of 0.4, flowering with an LAI max and harvesting with an LAI <0.4) and a finer calendar of the irrigation constraints based on expert opinion. The article is limited to the case of irrigated maize in the hills of Gascony. The article clearly

presents the approach and an improvement is observed, mainly on irrigation volumes, but this at the cost of determining 3 parameters based on the phenology of corn and 2 on the equipment (dose provided and minimum return time). The model is also very sensitive to soil characteristics such as AWCmax which is linked to soil depth. Finally, we are quite close to deterministic situations considered too complex and genericity in parameter determination is far to be clear (the authors must give 2 sets of parameters according to the plots studied, all located in the same sector). Moreover, if the improvement is clear in 2013, it is much less obvious in 2014. The article lacks a real discussion on the method to be used to generalize the approach in a context where there is a diversity of crops, cultivars for a given crop, soils, regional contexts and a diversity of equipment. That is a lot and this point should be the subject of much deeper discussion on what sources can be mobilized at the regional level and how they can be exploited. For example, information at the level of that provided by the FAO could ultimately be appropriate as it is a level at which we can synthetize knowledge and compile data. I suggest to consider in the discussion the following points :

- How to collect expert data on irrigation strategies and technical data. The authors suggest that the model might be calibrated on irrigation records. What would be the data sources then? Such an approach, which will probably be necessary, could have been tested on the data sets used by the authors.

- The quality of phenology determination is insufficiently discussed. What would be the impact of the absence of satellite data on phenological accuracy and the resulting impact on irrigation estimation? This can be addressed by a sensitivity analysis.

- AWCmax is an important factor in the proposed model. This one will never be known with certainty. Therefore, would it be possible to propose a single value (or a limited number if it appeared that the soil properties were a proven fact for the irrigation decision). What impact would this have on the estimation of irrigation?

- The accuracy with which the model simulates the soil water content at the beginning

of the irrigation period is not discussed. However, mistakes of several tens of mm can quickly be made.

- The model seems to be designed to run at a mesh resolution of 1 km. At this scale there will surely be similar crops with phenological stage differences. How will these discrepancies be managed?

- To properly cover a territory, how many types of crops will have to be considered?

To feed a model that should work on a 1km scale, it is as important to address these issues as to develop an algorithm that is flexible but whose parameters seem to be case dependent. As a result, we find ourselves in the same situation as explicit models (such as STICS as mentioned in the introduction and criticized). I therefore propose to give a much larger room to discussion and give concrete and argued suggestions for generalization. From a more specific point of view, here are some remarks on the text, which is generally very well read.

P2 L27-32 I am not sure that the proposed approach overcome such limitations P3 L15-20 I am not sure that the proposed approach is simpler. The cost of the flexibility is the number of parameters and thus it raises the problem of their determination. P3 L23-25 : it could be the appropriate level to determine parameters over large territories. P6 L28 contrarily to of all (spelling?) P7 L7 : is plot refer to field used in the study? P7L18 : I am surprised by the LAI definition. In general it the whole green area and in Bvnet I think that the training data set is related to whole leaf area. P9L20 in order to optimize the water resources availability (in order to optimize irrigation rate according to water resource availability?). P10L15 : 8 h looks very long. In general farmer use irrigation equipment which is a moving system where a unitary surface "see" irrigation during a much shorter time (30-45'). Setting an 8 hours irrigation will lead to a very small irrigation flux. If such a flux has importance in ISBA, I suggest to reduce it accordingly, even 8 hours is needed to irrigated the whole field.

P10 L28 : I am not sure to fully understand on which timr period Min and max are

established (at the annuel level LAImin=0) P12L28-29 : what consistent means here? Figure 2 : it is difficult to identify irrigation since the lines barely reach the threshold line..

---

## Referee Comment (RC2) · Anonymous Referee #2 · 16 May 2019

The paper describes an attempt to improve the modelling of irrigation in the Land Surface Model SURFEX-ISBA. The method used in the study aimed in particular in improving the simulation of timing and amount of irrigation by considering high resolution remote sensing imagery. Objective has been to make more realistic simulations possible while keeping the approach generic enough to enable global scale simulations. The topic fits very well to the scope of the journal and in general, the article is well written. However, I cannot recommend publication of the present version of the article in HESS. My major points of criticisms are:

1.) One essential improvement mentioned several times by the authors very promi-

nently is the consideration of the Leaf Area Index (LAI) detected by processing high resolution satellite imagery from several sensors. However, the authors miss completely to describe why this is needed and why using LAI is better than using other vegetation indices that are easier to calculate. The authors used imagery with a high spatial resolution (10 - 20 m) and a high revisiting time 3-5 days (page 8, line 5). With the help of the neural network tool BV-NET they derive LAI from reflectance values. After that, they averaged the derived LAI spatially for the plots compared in the study and also averaged the values in time to derive monthly values used in the LSM. I don't understand why such an effort is made when later the data will be averaged. Furthermore, deriving LAI from reflectance requires to know about the characteristics of the crop grown in the field (canopy architecture, leaf angle, crop height). This information is not available for large scale studies and using standard crop parameters introduces a considerable uncertainty into the LAI calculations. I'm therefore not convinced that using the LAI results in any improvement compared to the use of other vegetation indices that can be much easier computed with lower uncertainty such as NDVI or EVI.

2.) I doubt that the methods used by the authors are appropriate for large scale application of the model. Many characteristics described by the authors, for example in section 3.1 (LAI when irrigation of maize starts, end of the irrigation period 45 days before harvest, irrigation rate, minimal return time), are only representative for maize grown in Southern France and I don't see any way to gather this information for other regions and other crops. Therefore I doubt that the approach is generic enough for global scale applications, an objective postulated by the authors.

3.) Timing of irrigation and irrigation volume is derived in the present study mainly by considering LAI dynamics and the simulated actual soil moisture content. The authors show that using variable thresholds for the soil moisture to trigger irrigation events results in more realistic irrigation amounts for the region studied. However, again I'm not convinced that this finding can be generalized. In many other regions the timing of irrigation is fix and just determined by the water rights of the farmer. Furthermore, when

irrigation water is free of cost and provided by big irrigation canals, farmers do not have any motivation to save water and consequently, they will use all the water that is supplied. This is completely different from situations where farmers pump their own water. Here farmers are more flexible but have to pay for energy and therefore tend to use less water. So my general impression is that the authors managed very well to adjust the model to better reflect the specific situation in the region which they studied and to improve thereby the accuracy of the model results. However, this is on the expense of more complexity and an increasing number of assumptions and parameters. I don't see how the authors can manage to derive and implement this background knowledge at global scale. Consequently, because of these limitations, I see the risk that the authors turned their global scale LSM into a more detailed model that can only be applied successfully at regional level when all the background information is available.

Specific comments: 1) The figure captions are to general. In any case abbreviations used in the legends should be explained.

---

## Author Comment (AC1) · 20 May 2019

Dear Referee,

We would like to thank you for your review. Your comments are very useful and stress that the main objective of our study has not been sufficiently well explained. The limitations of our work have also not been sufficiently discussed. Please find hereafter a detailed answer to your comments.

"The paper proposes the development of an irrigation module in LSM that better reflects the reality of decisions that depend on the crop, its phenological stage and irrigation

techniques."

The main point of the paper is not the development of the module itself, as it is adapted from an existing irrigation module in models like SAFYE or SAMIR cited in the article. The objective is to present a methodology to ameliorate irrigation simulation at landscape scale by combining the advantages of high-resolution remote sensing and expertise from agricultural management agencies. We propose to better highlight this point.

"an improvement is observed, mainly on irrigation volumes, but this at the cost of determining 3 parameters based on the phenology of corn and 2 on the equipment (dose provided and minimum return time)( . . .)To feed a model that should work on a 1km scale, it is as important to address these issues as to develop an algorithm that is flexible but whose parameters seem to be case dependent. As a result, we find ourselves in the same situation as explicit models (such as STICS as mentioned in the introduction and criticized)."

Thank you for this comment. It seems that the distinction between the module development and the method to determine the parameters has been insufficiently explained. In fact, most of the LSMs, including ISBA, already use such parameters: irrigation period, dose, minimal return time, application time and triggering threshold. They are just fixed to climatological values or determined from other variables in the model. The purpose of our development is to give these parameters a liberty degree for users, allowing using the default climatological values as well as more precise information if available. As for the complexity of agronomical models such as STICS, it uses a stress function to trigger irrigation which does not depend only on the soil moisture but also on physiological parameters inherent to the crop simulated, which are not present in LSMs. Even if irrigation is simulated in a pretty similar way, spatializing such models is way more complex given the number of inputs to use.

As you point out, the lack of genericity lies in the method for determining the parameters:

"Finally, we are quite close to deterministic situations considered too complex and genericity in parameter determination is far to be clear (the authors must give 2 sets of parameters according to the plots studied, all located in the same sector). (. . .)The article lacks a real discussion on the method to be used to generalize the approach in a context where there is a diversity of crops, cultivars for a given crop, soils, regional contexts and a diversity of equipment. That is a lot and this point should be the subject of much deeper discussion on what sources can be mobilized at the regional level and how they can be exploited. For example, information at the level of that provided by the FAO could ultimately be appropriate as it is a level at which we can synthetize knowledge and compile data."

As mentioned above, our goal is to present a methodology to better simulate irrigation practices at landscape scale, essentially for water management purpose. We aim to show that using high-resolution remote sensing to spatialize theoretical irrigation practices is of great interest at this scale. Acquiring such expertise on irrigation on each crops and equipment is the main limitation of the method proposed and has not been discussed enough. At landscape scale, such information can be obtained from agricultural and water management agencies, like the CACG, farm cooperatives or government services. Such agencies exist in most agricultural regions and monitor many types of crops. Using their information allows adapting the method to the diversity of crops and practices. Such work still needs to be done. However, uncertainties remain on localizing actual irrigated plots, equipment types and, obviously, on farmers practices whose do not follow typical irrigation.

"The authors suggest that the model might be calibrated on irrigation records. What would be the data sources then? Such an approach, which will probably be necessary, could have been tested on the data sets used by the authors."

We did not perform calibration ourselves because we adopt expertise values for the

triggering thresholds at different phenological stages. The calibration is recommended when this expertise is not available. Data such as location, irrigation type, irrigation dates and volumes should be used for model calibration. The agencies mentioned above gather such information on several plots and can provide it in the scope of optimizing water use. The trend in data policy is also the open data. Another perspective should be to calibrate irrigation on high spatial and temporal resolution surface soil moisture products from micro-wave remote sensing such as Sentinel-1.

"Moreover, if the improvement is clear in 2013, it is much less obvious in 2014."

As the irrigation volumes are very small in 2014 (wet year), improvement is less clear than in 2013. As pointed out in discussion section, it denotes that in wet years, modeling irrigation is most challenging. But it is less crucial as there is a way smaller pressure on the water resources. It would be great to perform the same simulations on a longer time span but we lack of data to do it yet.

"The quality of phenology determination is insufficiently discussed. What would be the impact of the absence of satellite data on phenological accuracy and the resulting impact on irrigation estimation? This can be addressed by a sensitivity analysis."

Thank you for this relevant comment. Monthly values of LAI, determined from more frequent acquisitions, are used in our study and so is the triggering threshold. As vegetation indices varies quite slowly in time, compared to meteorological variables for example, linear interpolation between satellite acquisitions and monthly averaging is quite reasonable. With a multi-sensor approach like in this study, there is very little chance that we entirely miss the phenological cycle. It could lead to differences of one or two weeks on the beginning and ending of the irrigation period. However, the difference between STD and STD-SAT in the manuscript shows there seems to be little impact on the irrigation estimation. The recent satellites with high revisit frequency like Sentinel-2 or Ven$\mu$s, contributes to attenuate this problem even more. If the missing rate of remote sensing data is too high, climatological values could also be used, with

the uncertainties associated (Etchanchu et al., 2017).

"The model is also very sensitive to soil characteristics such as AWCmax which is linked to soil depth. (. . .)AWCmax is an important factor in the proposed model. This one will never be known with certainty. Therefore, would it be possible to propose a single value (or a limited number if it appeared that the soil properties were a proven fact for the irrigation decision). What impact would this have on the estimation of irrigation?"

Every irrigation scheme piloted by soil moisture thresholds, like in the vast majority of LMSs, is very sensitive to AWCmax. Proposing a single value for maximal rooting depth on each crop type is possible given the fact that farmers often do not know if their soil depth is limiting the root development or not. The uncertainty to be done in this case is presented in section 4.1.2 and 5.2, particularly in fig. 5 and 6. The analysis shows that it is better to overestimate the rooting depth than underestimating it.

"The accuracy with which the model simulates the soil water content at the beginning of the irrigation period is not discussed. However, mistakes of several tens of mm can quickly be made."

The uncertainty on this is hard to estimate because of the thresholds effects. Small differences in initial soil moisture may affect or not the annual irrigation volume depending on the miss or the addition of an irrigation event due to reaching the thresholds at slightly different timings. In any case, the timing of irrigation would be modified depending on the climatic conditions. In dry conditions, the soil moisture drops sufficiently fast to create few differences in irrigation timing (maybe few days). In wet conditions, the timing may be impacted way more but as annual irrigation volume is smaller, the difference on it should be no greater than one or two irrigation events.

"The model seems to be designed to run at a mesh resolution of 1 km. At this scale there will surely be similar crops with phenological stage differences. How will these discrepancies be managed?"

This study is done in the scope of ECOCLIMAP-SG, the land cover parameters map which will replace ECOCLIMAP-II in future versions of SURFEX-ISBA. This database describes the land cover at 300m resolution and the computation cell will have a single vegetation type. A 300m resolution could limit the discrepancies in phenology in a same computation cell. By the way, our previous study (Etchanchu et al., 2017) shows the interest in simulating at landscape scale with homogeneous plots as computation cells and we aim at using such approach to simulate irrigation impact at the same scale.

"To properly cover a territory, how many types of crops will have to be considered?"

Given the plant functional types of SURFEX, 6 different types of vegetation are considered as possibly irrigated in simulations: winter C3 crops, summer C3 crops, C4 crops, temperate broadleaf deciduous (fruit trees), temperate broadleaf evergreen (evergreen fruit trees, like olive trees) and shrubs.

Please find below answers to your specific comments :

"P2 L27-32 I am not sure that the proposed approach overcomes such limitations"

As the irrigation dose can be forced and modulated in space and time, it can be adapted to the practices, contrarily to the models which calculate the dose from the soil moisture. The main point is to get the information. However, some LSMs already use a fixed dose based on climatology.

"P3 L15-20 I am not sure that the proposed approach is simpler. The cost of the flexibility is the number of parameters and thus it raises the problem of their determination."

As said earlier, the flexibility does not increase the number of parameters. It just allows the users to determine them with spatiotemporal variations if accurate information is available, instead of using climatological values.

"P3 L23-25: it could be the appropriate level to determine parameters over large territories."

That is right for regional to global scale.

"P7 L7 : is plot refer to field used in the study?"

Yes plot refers to field in this study. You could advise us on the term you think is the best fit in this context as we are not English native speakers and we will harmonize the vocabulary accordingly.

"P7L18: I am surprised by the LAI definition. In general it the whole green area and in Bvnet I think that the training data set is related to whole leaf area."

The definition is directly taken from Chen and Black (1992). It is justified because the whole green area includes both the above and under sides of the leaves whereas only the above is photosynthecally active, explaining why we take only half of the green leaves area. The training dataset of BVNET is based on this definition.

"P9L20 in order to optimize the water resources availability (in order to optimize irrigation rate according to water resource availability?)."

It is more justified by the fact that the plants have very low water consumption as their evapotranspiration is low before this stage. Therefore, it is generally not necessary to irrigate before this stage, except in case of drought, when irrigation may be used to trigger the emergence of the plants.

"P10L15 : 8 h looks very long. In general farmer use irrigation equipment which is a moving system where a unitary surface "see" irrigation during a much shorter time (30-45'). Setting an 8 hours irrigation will lead to a very small irrigation flux. If such a flux has importance in ISBA, I suggest to reduce it accordingly, even 8 hours is needed to irrigated the whole field."

You are right but in reality, the entire plot is not irrigated at the same time. Thus the runoff generated on the surface goes to other parts of the plot. Using such irrigation application time could generate unrealistic runoff over the entire plot. The 8 hours value is linked to the position change frequency of irrigation reels, mostly used in the plots

simulated. It could also be appropriate for pivot irrigation or fixed spray irrigation. Note that native version in ISBA applied irrigation dose on a unique time step which may be unrealistic depending on the time step chosen.

"P10 L28 : I am not sure to fully understand on which time period Min and max are established (at the annuel level LAImin=0)"

The minimal value is determined at the annual level as it is reached after the harvest or before the plant emergence. Its value is not necessarily equal to zero because BVNET do not simulate easily a null LAI. As for the maximal value, we used a filtering period to determine it, from May the 1st to the end of the year, in order to avoid confusions with intermediate winter crops. We will precise it in the revised version.

"P12L28-29 : what consistent means here?"

It means that the scores do not vary in great proportions between years for the FIXE and VARI experiments, at least concerning the annual irrigation volumes. Doing the simulation on a longer time span may confirm the hypothesis that this parameterization is more stable in time than the one in STD and STD-SAT.

"Figure 2 : it is difficult to identify irrigation since the lines barely reach the threshold line.."

We will rework this figure for the revised version of the manuscript.

We hope the answers given meet your expectations. If any other question arises, we will be glad to answer it.

---

## Author Comment (AC2) · 22 May 2019

Dear Referee,

We would like to thank you for your review. It mainly points out that we have not been clear enough on the scope of the study (water management at the landscape scale) and opened too much the discussion on the global scale. Please find hereafter answers to your comments:

"1.) One essential improvement mentioned several times by the authors very prominently is the consideration of the Leaf Area Index (LAI) detected by processing high

resolution satellite imagery from several sensors. However, the authors miss completely to describe why this is needed and why using LAI is better than using other vegetation indices that are easier to calculate."

We used LAI for 2 reasons. Firstly, SURFEX-ISBA uses LAI as input, like most LSMs, which assumed this variable to be representative of the transpiring area of the plants. Determining LAI instead of simpler vegetation indices is thus necessary. Secondly, we have reference values about measured LAI at the phenological stages mentioned in the study. As the LAI was necessary for computational reasons, it was natural for us to also take it to determine irrigation rules. But we agree that simpler vegetation indices could be good drivers of surfaces processes, such as ETR or Photosynthesis.

"The authors used imagery with a high spatial resolution (10 - 20 m) and a high revisiting time 3-5 days (page 8, line 5). With the help of the neural network tool BV-NET they derive LAI from reflectance values. After that, they averaged the derived LAI spatially for the plots compared in the study and also averaged the values in time to derive monthly values used in the LSM. I don't understand why such an effort is made when later the data will be averaged."

The LAI is averaged spatially at the field scale, which is here considered homogeneous, to save computation time and keep the interest of the high resolution by remaining on cells of simulations composed of a single species (opposed to mixed grid cell), as presented in our previous study (Etchanchu et al., 2017) In addition, the plot is the irrigation management unit. The temporal averaging is done essentially for technical reasons. SURFEX simulations are heavier to perform with daily forcing of the surface parameters. Simulations with decadal forcing could be envisaged. But the study mentioned previously also showed that monthly forcing of LAI is sufficient to increase significantly the simulated fluxes accuracy compared to using climatologies.

"Furthermore, deriving LAI from reflectance requires to know about the characteristics of the crop grown in the field (canopy architecture, leaf angle, crop height). This

information is not available for large scale studies and using standard crop parameters introduces a considerable uncertainty into the LAI calculations. I'm therefore not convinced that using the LAI results in any improvement compared to the use of other vegetation indices that can be much easier computed with lower uncertainty such as NDVI or EVI."

As written above, the LAI is necessary for SURFEX. LAI calculated with BVNET is one of the best products already available on our study area and that is why we use it. By the way, new Copernicus products offer dynamic LAI at global scale at a resolution of 300m (https://land.copernicus.eu/global/products/lai).

"2.) I doubt that the methods used by the authors are appropriate for large scale application of the model. Many characteristics described by the authors, for example in section 3.1 (LAI when irrigation of maize starts, end of the irrigation period 45 days before harvest, irrigation rate, minimal return time), are only representative for maize grown in Southern France and I don't see any way to gather this information for other regions and other crops. Therefore I doubt that the approach is generic enough for global scale applications, an objective postulated by the authors."

We have not been clear enough on the scope of our study and are sorry for this. Global scale application is not an objective of this study. We aim at proposing ways to better simulate irrigation at landscape scale, essentially for water management applications. At this scale, agronomical models are too complex as they need many more inputs than Land Surface Models. The irrigation scheme development presented in the study only consists in giving a liberty degree on fixed irrigation parameters already present in the model. Determining values other than climatological values can be achieved by exploiting both technical documentation on crop management, given by government or private services in charge of agricultural management in the simulated area, and remote sensing data. The remote sensing data allows applying theoretical irrigation practices, which often depends on crop phenological stages, with the actual phenology of the crops.

[Figure]

"3.) Timing of irrigation and irrigation volume is derived in the present study mainly by considering LAI dynamics and the simulated actual soil moisture content. The authors show that using variable thresholds for the soil moisture to trigger irrigation events results in more realistic irrigation amounts for the region studied. However, again I'm not convinced that this finding can be generalized. In many other regions the timing of irrigation is fix and just determined by the water rights of the farmer. Furthermore, when irrigation water is free of cost and provided by big irrigation canals, farmers do not have any motivation to save water and consequently, they will use all the water that is supplied. This is completely different from situations where farmers pump their own water. Here farmers are more flexible but have to pay for energy and therefore tend to use less water."

Many agricultural areas in the world face or will face water management issues, especially in a changing climate context. This study shows that for such landscape scale applications, instead of taking climatological values for the irrigation parameters, expertise from agricultural management agencies can be used and spatialized by the mean of remote sensing data, leading in a more realistic estimation of the irrigation.

"So my general impression is that the authors managed very well to adjust the model to better reflect the specific situation in the region which they studied and to improve thereby the accuracy of the model results. However, this is on the expense of more complexity and an increasing number of assumptions and parameters. I don't see how the authors can manage to derive and implement this background knowledge at global scale. Consequently, because of these limitations, I see the risk that the authors turned their global scale LSM into a more detailed model that can only be applied successfully at regional level when all the background information is available."

As mentioned above, the number of parameters of the irrigation scheme is not increased as all the parameters already exist in most LSMs: irrigation period, dose, minimal return time, application time, triggering threshold. They are determined on the form of fixed climatological values or from other model variables. We just give them a

liberty degree. This way, users can keep using default values but can also use more precise information if available. The particular values presented in this study are indeed adapted to maize crops. But the general methodology can be applied on every agricultural area where technical documentation about crop irrigation management exists. Such information is increasingly available on many regions. On regions with lack of such documentation, calibration should be performed instead of making assumptions on the triggering threshold. The data to calibrate on would be irrigation dates and dose. Such data can be obtained by questioning farmers, monitoring several representative plots in agricultural areas or from agencies in charge of agricultural management in the area. Calibration on surface soil moisture from high spatiotemporal resolution micro-wave remote sensing products, such as the surface soil moisture derived from Sentinel-1 data, could also be envisaged. Global scale simulation is not the scope of the present study but we aim at presenting a methodology to increase irrigation simulation accuracy at landscape scale with the help of high resolution remote sensing.

We hope the answers given meet your expectations. If any other question arises, we will be glad to answer it.